# Antifungal Activities of *cis*-*trans* Citral Isomers against *Trichophyton rubrum* with ERG6 as a Potential Target

**DOI:** 10.3390/molecules26144263

**Published:** 2021-07-14

**Authors:** Yin Zheng, Yanhong Shang, Mengyun Li, Yunzhou Li, Wuqing Ouyang

**Affiliations:** 1College of Animal Science, Guizhou University, Guiyang 550025, China; 2College of Animal Science and Veterinary Medicine, Henan Agricultural University, Zhengzhou 450002, China; shangyh1979@gmail.com; 3College of Animal Science and Technology, Henan University of Science and Technology, Luoyang 471023, China; mengyun.li@163.com; 4College of Agriculture Science, Guizhou University, Guiyang 550025, China; liyunzhou2007@126.com; 5College of Veterinary Science, Northwest A&F University, Yangling 712100, China; wqoy1960@outlook.com

**Keywords:** neral, geranial, *Trichophyton rubrum*, ergosterol biosynthesis, ERG6, citral, antifungal activity, dermatophytes

## Abstract

*Trichophyton rubrum* causes ringworm worldwide. Citral (CIT), extracted from *Pectis* plants, is a monoterpene and naturally composed of geometric isomers neral (*cis*-citral) and geranial (*trans*-citral). CIT has promising antifungal activities and ergosterol biosynthesis inhibition effects against several pathogenic fungi. However, no study has focused on neral and geranial against *T. rubrum*, which hinders the clinical application of CIT. This study aimed to compare antifungal activities of neral and geranial and preliminarily elucidate their ergosterol biosynthesis inhibition mechanism against *T. rubrum*. Herein, the disc diffusion assays, cellular leakage measurement, flow cytometry, SEM/TEM observation, sterol quantification, and sterol pattern change analyses were employed. The results showed geranial exhibited larger inhibition zones (*p* < 0.01 or 0.05), higher cellular leakage rates (*p* < 0.01), increased conidia with damaged membranes (*p* < 0.01) within 24 h, more distinct shriveled mycelium in SEM, prominent cellular material leakage, membrane damage, and morphological changes in TEM. Furthermore, geranial possessed more promising ergosterol biosynthesis inhibition effects than neral, and both induced the synthesis of 7-Dehydrodesmosterol and Cholesta-5,7,22,24-tetraen-3β-ol, which represented marker sterols when ERG6 was affected. These results suggest geranial is more potent than neral against *T. rubrum*, and both inhibit ergosterol biosynthesis by affecting ERG6.

## 1. Introduction

Dermatophytosis is caused by a group of dermatophytes, among which *Trichophyton rubrum* is the most prevalent worldwide. *T**. rubrum* infection is commonly recognized as tineas in the skin and severely impairs the welfare of infected mammals [1]. At present, controlling dermatophytosis primarily relies on available antifungals (such as fluconazole, ketoconazole, and terbinafine) [1,2]. However, their side effects and decreased sensitivity lead to unsatisfactory treatment outcomes; therefore, alternatives to current antifungals are required [3]. Essential oils, extracted from medicinal plants, are composed of aromatic compounds (both polar and non-polar) and are widely used in embalming processes and folk medicine due to their promising antimicrobial activities [3,4,5]. In the last decade, several essential oils that are unlikely to cause drug resistance have been reported to display promising antifungal activities, and they may remedy drug sensitivity to conventional antifungals [5,6]. 

Citral (CIT), extracted from plants of the genus *Pectis*, a monoterpene, is naturally composed of two geometric isomers, neral (*cis*-citral) and geranial (*trans*-citral), which exhibit promising effects against *Trichophyton* spp. [7,8] and *Candida albicans* [9,10], as well as other pathogenic fungi in vitro [11,12,13,14]. Several CIT antifungal mechanisms have been reported, including cell wall damage [13], mitochondrial membrane potential disruption [12], and membrane damage [9,14]. However, no further research has been conducted to compare and elucidate neral and geranial activities and their ergosterol biosynthesis inhibition mechanism, which has hindered the clinical application of CIT in treating dermatophytosis. 

Given the above, first, this study aimed to compare neral and geranial activities and then preliminarily elucidate their ergosterol biosynthesis inhibition mechanism against *T. rubrum*. *Trichophyton rubrum* ATCC28188 was used as the test strain.

## 2. Results and Discussion

CIT is naturally composed of geometric isomers (*cis*-neral and *trans*-geranial). The MIC and MFC of neral against *T. rubrum* were 111.23 μg/mL and 222.45 μg/mL, respectively, while that of geranial were 55.61 μg/mL and 111.23 μg/mL, respectively. This implies that geranial is more potent against *T. rubrum* than neral. Similar results were observed in studies involving *Aspergillus flavus* [14] and *Candida albicans* [9]. In a checkerboard test, results were obtained in terms of MIC, where neral was 111.23 μg/mL and geranial was 0.87 μg/mL, or neral 1.74 μg/mL and geranial 55.61 μg/mL; therefore, FICI was 1.016, and, thus, these two isomers showed no interaction with each other, indicating that they can be used in combination.

The disc diffusion assay is easy to perform and is commonly used for drug susceptibility against yeasts [15] and dermatophytes [3]. Here, disc diffusion was used for its data visualization feature to vividly illustrate and compare neral and geranial antifungal effects, where TB was used as a drug control with MIC 0.016 μg/mL and MFC 0.032 μg/mL. The results are presented in Table 1 and Figure 1. Geranial inhibition zones (Figure 1C) were significantly greater in diameter than neral (Figure 1D) (*p* < 0.01) in terms of MIC and MFC, and comparable to TB (Figure 1B) (*p* > 0.05). These results implied that geranial was more potent than neral against *T. rubrum*. 

SEM and TEM have been widely applied in pharmacological science to investigate antifungal mechanisms of unknown compounds quickly and vividly [11]. SEM was used to observe mycelium morphology and cell wall integrity after treatment with agents. The control showed characteristic mycelial growth with smooth and integrated mycelial walls (Figure 2A), while the MIC_TB_ group (Figure 2B) exhibited sunken and shriveled mycelia with integrated mycelial walls. MIC_Neral_ and MIC_Geranial_ groups (Figure 2C,D) both exhibited integrated but twisted mycelial walls, and the MIC_Geranial_ group displayed some shriveled mycelia (Figure 2D). It has been reported that CIT can induce cell membrane damage and intercellular material leakage in other pathogenic fungi, resulting in mycelium deformation [7,9]. The SEM results were further confirmed by TEM images of conidia treated with specific treatments, where the control exhibited clear and integrated conidia structures (Figure 2E), with many mitochondria and ribosomes scattered in the cytoplasm and a large number of vacuoles, indicating distinct membrane damage and cellular leakage (Figure 2F), consistent with the membrane-damaging antifungal mechanism. Conidia in MIC_Neral_ and MIC_Geranial_ showed cell membrane damage, cellular content leakage, swelling vacuoles, polysaccharide particles, and distorted mitochondria (Figure 2G,H). These changes may be partly due to the following reasons: (i) swollen vacuoles represent a fungal protective mechanism, whereby the fungi store and prevent antifungal agents from contacting intercellular organelles [16]; (ii) polysaccharide particles are derived from glycoproteins of damaged cell membranes [17]; (iii) distorted mitochondria, indicating fungal death, appear immediately after the mitochondrial transmembrane potential changes caused by intercellular material leakage [18]. The SEM/TEM images suggested that neral and geranial could inhibit *T. rubrum* by inducing membrane damage. 

Cellular leakage measurement is used to quantitatively and vividly illustrate cellular material leakage due to membrane damage; however, cell wall damage can induce osmotic pressure changes, which result in membrane damage [19]. To verify whether membrane damage occurs due to neral and geranial, sorbitol, an osmotic protectant that effectively stabilizes the fungal protoplast [20], was added into the MOPS buffer. Therefore, agents resisting fungal cell wall damage would exhibit reduced cellular leakage rates, while samples with membrane-damaging agents would display significant cellular leakage. The results (Figure 3) showed that cellular leakage rates significantly decreased in the MFC_CA_ group (*p* < 0.01) after 0.8 M sorbitol was added to the MOPS buffer, guaranteeing the validity of the results, while no significant changes (*p* > 0.05) were observed in the MIC_Neral_ or MIC_Geranial_ groups before and after sorbitol was added, inconsistent with the cell wall damage mechanism of action of neral and geranial. It is worth noting that MIC_Geranial_ displayed significantly higher (*p* < 0.05) cellular leakage rates than MIC_Neral_ at all time intervals with or without sorbitol, suggesting that geranial was more potent than neral against *T. rubrum*. Moreover, these results also indicated neral and geranial antifungal mechanisms against *T. rubrum*, including membrane integrity disruption, but no cell wall damage. Similar results were also observed in reports involving *A**. flavus* [18] and *C**. albicans* [9].

We employed flow cytometry and PI staining to further quantitively compare the membrane-damaging effects of neral and geranial on *T. rubrum*. Conidia with damaged membrane abnormally internalize PI by binding to nuclear DNA, resulting in increased red fluorescence that can be detected using flow cytometry [21,22]. PI red fluorescence emissions appeared significantly greater in each treatment (Figure 4B,F: MIC_TB_; C,G: MIC_Neral_; D,H: MIC_Geranial_; Figure 5) (*p* < 0.01) compared to controls (Figure 4A,E and Figure 5). At all time intervals, the greatest PI-positive emissions were observed in the MIC_TB_ group, followed by MIC_Geranial_ and MIC_Neral_, where MIC_Geranial_ appeared significantly greater than MIC_Neral_ (*p* < 0.01 or 0.05). These results indicate that neral and geranial may disrupt cell membrane integrity and that geranial exerts a more potent membrane damaging effect.

Ergosterol is necessary for maintaining cell membrane integrity and permeability; its abnormal reduction or absence leads to cellular leakage resulting in fungal death, and its synthetic process is a potential drug target for the development of novel antifungal agents [21,23]. We performed a sterol quantification test to confirm whether neral or geranial inhibits ergosterol biosynthesis; the results are listed in Figure 6. After treatment for 24 h, ergosterol levels in MIC_Neral_, MIC_Geranial_, and MIC_TB_ were significantly (*p* < 0.01, *p* < 0.05) decreased; in contrast, the amount of late sterol intermediate 24(28) was significantly (*p* < 0.01, *p* < 0.05) increased when compared with that of the control, indicating the former intermediate sterols in the ergosterol biosynthesis pathway were accumulated instead of becoming ergosterol [24]. In addition, geranial showed more pronounced changes than neral (*p* < 0.05, *p* < 0.01). These results indicate that neral and geranial can inhibit ergosterol biosynthesis and cause late 24(28) dehydroergosterol (DHE) accumulation. 

At present, several antifungals (azoles, allylamines, and morpholines), which exert their antifungal activities by inhibiting specific enzymes that regulate ergosterol biosynthesis, are used to treat dermatophytosis [22,23]. A total of eight distinct enzymes participate in the conversion of lanosterol (the sterol) to ergosterol; however, most of these enzymes are membrane associated and unstable after isolation [23,24]. Further, target enzyme validation requires gene deletion experiments, which is a great challenge because only rudimentary protocols are presently available [25,26], compromising the development of novel ergosterol synthesis inhibitors. Recently, several studies reported that sterol pattern changes could be used as direct evidence to indicate that specific ergosterol biosynthesis enzymes are affected in several pathogenic fungi [27,28], which facilitate the identification of target enzymes after treatment with ergosterol synthesis inhibitors. In this assay, a total of 15 sterols were detected and analyzed; specifically, 12 in controls, 10 in MIC_TB_, and 15 in MIC_Neral_ and MIC_Geranial_. Changes in each specific sterol (Table 2) and gas chromatograms are shown in Figure 7. It has been reported that it is relatively easy to insert monoterpenoids into fungal cell membranes to disrupt ergosterol synthesis. Our results (Table 2 and Figure 7) showed that zymosterol and lanosterol, former sterols catalyzed by C24-methyltransferase (ERG6) into fecosterol and eburicol [23], respectively, are abnormally accumulated after treatment with MIC_Neral_ and MIC_Geranial__._ Meanwhile, it is worth noting that 7-Dehydrodesmosterol, Cholesta-7,24-dien-3β-ol, and Cholesta-5,7,22,24-tetraen-3β-ol were only detectable after treatment with MIC_Neral_ and MIC_Geranial_, but undetectable in controls or MIC_TB_. According to the main ergosterol biosynthesis pathways under enzyme inhibition [23,27], 7-Dehydrodesmosterol and Cholesta-5,7,22,24-tetraen-3β-ol represent marker sterols when ERG6 is inhibited. These results imply that (i) ERG6 is the target enzyme for neral and geranial and that (ii) geranial more potently affects ERG6 by inducing a greater accumulation of zymosterol and lanosterol than neral. However, it should be emphasized that the results here can only indicate ERG6 as neral and geranial’s potential target enzyme. To further verify how ERG6 is affected, gene deletion and specific enzyme inhibition experiments are still needed in the future. 

## 3. Materials and Methods

### 3.1. Test Strain and Chemical Compounds

*T. rubrum* ATCC 28188 was purchased from the American Type Culture Collection (ATCC). Neral and geranial were synthesized via enzymatic synthesis methods and confirmed using high-performance liquid chromatography, as reported by Luo et al. [14]. We obtained terbinafine (TB) and caspofungin (CA) from Jianglai Biotechnology Co., Ltd. (Shanghai, China), standard cholesterol (99% purity) and cholestane (98% purity) from Aladdin (Shanghai, China), sorbitol from Xilong Chemical Co., Ltd. (Chengdu, Sichuan), and 3-(*N*-morpholino)propanesulfonic acid (MOPS) from Solarbio Technology Co., Ltd. (Beijing, China).

### 3.2. Minimal Inhibitory Concentrations (MICs)/Minimal Fungicidal Concentrations (MFCs) Determination 

Initially, the *T. rubrum* conidial suspension was prepared using MOPS buffer and filtered through Whatman filter paper (pore size 11 μm) as to remove hyphal fragments, and then adjusted to 1.5 × 10^4^ CFU/mL. The broth macrodilution assay was conducted according to the methods recommended by CLSI (2008) [29] for filamentous fungi, and the MICs/MFCs of TB/CA/neral/geranial were determined. 

### 3.3. Interaction of Neral with Geranial

A checkerboard method was applied to evaluate the interaction of neral and geranial against *T. rubrum*. Briefly, the *T. rubrum* conidial suspension (1.5 × 10^5^ CFU/mL) was prepared as previously stated. Other procedures were adopted from the study by Khan et al. [22]. The interaction types were determined according to fraction inhibitory concentrations (FICs), which were calculated as the MICs of the combination of neral with geranial divided by the MICs of neral or geranial alone. FICI (FICI) was attained by adding both FICs. The results were explained as follows: FICI ≤ 0.5, synergistic; >0.5–4.0, no interaction; and >4.0, antagonistic as described by Odds (2003) [30].

### 3.4. Disc Diffusion Assay

The disc diffusion assay was conducted according to López-Oviedo et al. (2006) [15] for filamentous fungi. Briefly, the *T. rubrum* conidial suspension (1.5 × 10^5^ CFU/mL) was prepared. One hundred microliters of conidial suspension was spread evenly on SDA plates (diameter: 9 cm) and dried at room temperature for 30 min. After that, two sterile Whatman^TM^ filter paper discs (6 mm in diameter) were placed in each SDA plate. Fifteen microliters of TB/neral/geranial, 100-fold MIC/MFC, was added to specific discs, and discs containing 15 μL of MOPS buffer were used as controls. Finally, all plates were cultured at 28 ± 2 °C with 65% humidity for five days; the assays were performed in triplicate, and results are expressed as means ± standard deviation.

### 3.5. SEM/TEM Observations

We employed SEM methodology from Aljabre et al. [31] with modifications to observe hyphae. Briefly, the phosphate-buffered solution (PBS, 100 µL) containing *T. rubrum* conidia (1.5 × 10^5^ CFU/mL) was evenly spread on SDA plates (9 cm diameter). All plates were thoroughly dried (28 °C) and incubated for five days at 28 ± 2 °C with 65% humidity. At the end of the incubation, a 5 mm diameter mycelium plug was cut from the central area of each SDA plate. Thereafter, each plug was placed at the center of the SDA plate containing MIC_Neral_ and MIC_Geranial_, which were used as test groups, and MIC_TB_ and SDA (no antifungal agents), which served as drug control and control, respectively, and were incubated for five days at 28 ± 2 °C with 65% humidity. After incubation, a 5 mm diameter mycelium plug adjacent to the SDA center was cut from each SDA plate. The mycelium plug was processed according to SEM sample preparation protocols and viewed using a Hitachi-4800S microscope (10 kV) (Hitachi High-Technologies Corporation, Tokyo, Japan).

We then employed TEM to observe conidia. Ten milliliters RPMI-1640 medium containing conidia (1.5 × 10^3^ CFU/mL) and 0.001% (*v*/*v*) Tween 80 with 100× drug concentrations were added individually to obtain MIC_TB_, MIC_Neral_, and MIC_Geranial_, while a medium without antifungal agents was used as the control. Thereafter, media were cultured at 28 ± 2 °C with 65% humidity using a shaker (60 rpm). After incubation for 24 h, the conidia were collected via centrifugation (3000 rpm, 5 min), washed with PBS, and centrifuged three times. The precipitated conidia were gently poured into a 55 °C aqueous agar solution (*w*/*v*, 15%), cooled at room temperature, prepared as 1 mm^3^ cubes, and fixed in 4% glutaraldehyde for 12 h. Finally, the cubes were processed according to TEM sample preparation protocols and viewed using a JEM-100CXⅡ at 75 kV (Shimadzu Corporation, Kyoto, Japan). 

### 3.6. Cellular Leakage Measurement

The methods for cellular leakage measurement were adopted from Lunde et al. [32] with some modifications. Initially, the MOPS buffer with and without 0.8 M sorbitol was prepared. Thereafter, *T. rubrum* conidia suspensions (1.5 × 10^3^ CFU/mL) were prepared in MOPS buffer and received a specific 100-fold concentrated solution to obtain MIC_Neral_ or MIC_Geranial_, which served as the test groups. MOPS buffer with MIC_CA_ and without any antifungal agents served as controls, while conidia treated with alcoholic potassium hydroxide solution served as a 100% cellular leakage control. Conidia were cultured at 28 ± 2 °C with 65% humidity using a shaker (60 rpm). The supernatant was removed from each treatment at 8 h and 24 h intervals. The 2 mL aliquot of MOPS buffer was centrifuged at 8000 rpm for 5 min, and the supernatant was analyzed for 260 nm absorbing materials in the buffer. Cellular leakage rates were calculated as the percentage of cellular leakage to the 100% cellular leakage control. Results are represented as the mean ± standard deviation based on three independent tests.

### 3.7. Flow Cytometry

We employed the methodology of Khan et al. (2011) [22] with certain modifications. Initially, RPMI 1640 medium with and without 400 μg/mL exogenous ergosterol was prepared. Briefly, *T. rubrum* conidial suspensions (1.5 × 10^3^ CFU/mL) were prepared in RPMI 1640 medium and received neral/geranial to obtain the MIC test groups. Media without antifungal agents and MIC_TB_ served as controls. Conidia were incubated as stated above. We then prepared the samples for propidium iodide (PI) analysis. Briefly, 5 mL of conidial solution was centrifuged at 3000 rpm for 3 min, conidia were resuspended in 5 mL of PBS, and PI was added (final concentration: 1 μg/mL PI in the medium) to each sample at 8 h and 24 h intervals. Conidia were then incubated at 35 °C for 30 min in the dark. Following incubation, all samples were analyzed using an FACS-Calibur flow cytometer (Becton Dickinson Biosciences) using a blue argon laser at 488 nm and 15 mW in the FL1 channel for cell-associated fluorescence and red fluorescence at 650 nm (15 mW) in the FL2 channel. Both channels were recorded in logarithmic scales for a minimum of 10,000 events per sample. For data analysis, the fluorescence of each treatment was recorded in FL1 and FL2 channels simultaneously, and four quadrants were determined according to the density plots of fluorescence intensity in the drug-free control, where more than 98% of cells were in the lower left quadrant. The percentages of cells located in the upper-left quadrant from each treatment were compared with the drug-free controls. Results are represented as the mean ± standard deviation based on three independent tests.

### 3.8. Sterol Quantitation Test

Total intracellular sterols from *T. rubrum* were extracted as described by Arthington-Skaggs et al. (1999) [27]. Briefly, conidial suspensions were prepared as described above, treated with MIC_TB_, MIC_Neral_, MIC_Geranial_, and RPMI1640 medium without any antifungal agents (control), and incubated at 28 ± 2 °C with 65% humidity using a shaker (60 rpm) for five days. After incubation, the media were filtered through sterile filter paper (Whatman #1, pore size 9 μm) to collect mycelium/conidia, washed with ultrapure water three times, and dried overnight at −60 °C. Subsequently, according to the methods introduced by Khan et al. (2010) [6], samples (0.1 g) from each treatment were processed to extract sterols and analyzed using a Shimadzu spectrophotometer at wavelengths of 281.5 nm and 230 nm, respectively. Finally, ergosterol content was expressed as the percentage of the dry weight of mycelium/conidia using the following equations:%Ergosterol + %24(28)DHE = [(A281.5/290) × 5]/0.1,(1)
%24(28)DHE = [(A230/518) × 5]/0.1(2)
%Ergosterol = [%Ergosterol + %24(28)DHE] − %24(28)DHE(3)

The test was performed in triplicate, and the results are expressed as means ± standard deviation.

### 3.9. Sterols Pattern Analysis

Conidial suspensions were prepared as previously stated. MIC_Neral_ and MIC_Geranial_ served as test groups, while MIC_TB_ and the medium without antifungal agents served as controls. All groups were cultured as described above. For cell lysis, extraction of nonsaponifiable matter, derivatization, and GC-MS analysis of sterol pattern changes were adopted from Müller et al. (2017) [24], where cholestane and cholesterol were used as internal standard substances (IS). Finally, the peak areas of specific intermediate sterols were compared with controls, and the results are expressed as a mean value based on three independent tests.

## 4. Conclusions

Through serial assays and tests, the results indicated that (1) geranial is more potent than neral against *T. rubrum* and (2) they exert ergosterol biosynthesis inhibition with ERG6 as a potential target. The results suggest that neral and geranial are promising agents for the clinical control of *T. rubrum* infection in the future.

## Figures and Tables

**Figure 1 molecules-26-04263-f001:**
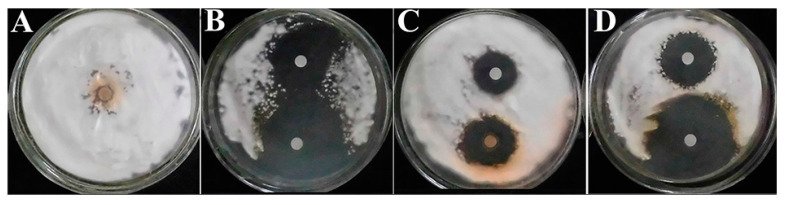
Disc diffusion test. (**A**–**D**) represent control, terbinafine, neral, and geranial treatments, respectively; the upper inhibition zones were formed by minimum inhibitory concentration, and the lower ones were formed by minimum fungicidal concentration.

**Figure 2 molecules-26-04263-f002:**
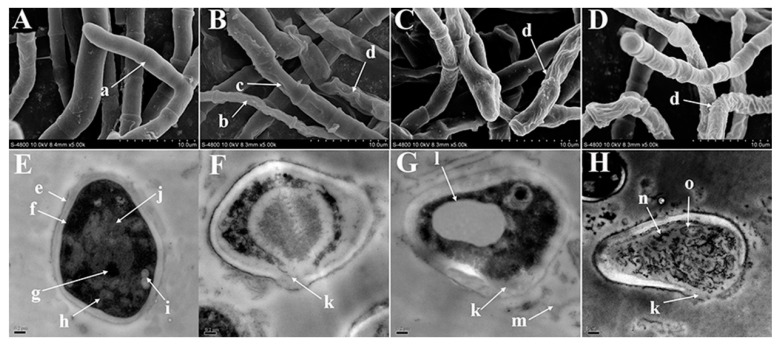
The SEM images of *Trichophyton*
*rubrum* mycelium and TEM images of *T. rubrum* conidia treated with specific agents for five days. (**A**,**E**) control; (**B**,**F**) terbinafine (MIC); (**C**,**G**) neral (MIC); (**D**,**H**) geranial (MIC). (a) distinct branch structure; (b) shriveled mycelium; (c) extreme wrinkles on the surface of mycelium; (d) twisted mycelium surface; (e) cell wall; (f) cell membrane; (g) cell nucleus; (h) mitochondria; (i) vacuole; (j) liposome; (k) broken cell membrane; (l) swollen vacuole; (m) intercellular material; (n) polysaccharide particles; (o) distorted mitochondria. MIC, minimum inhibitory concentration.

**Figure 3 molecules-26-04263-f003:**
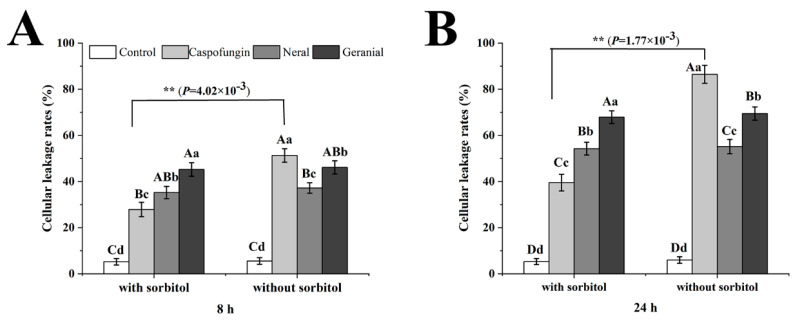
Cellular leakage rates of *Trichophyton rubrum* conidia treated with specific agents at minimum inhibitory concentration for 8 h (**A**) and 24 h (**B**). Within the minimum fungicidal concentration (MFC)_CA_ and same time intervals, “**” indicates a significant difference (*p <* 0.01) between those with sorbitol and without sorbitol; within the same medium (with sorbitol or without sorbitol) and time intervals, columns with lowercase or uppercase letters indicate significant or highly significant differences (*p* < 0.05 or *p* < 0.01).

**Figure 4 molecules-26-04263-f004:**
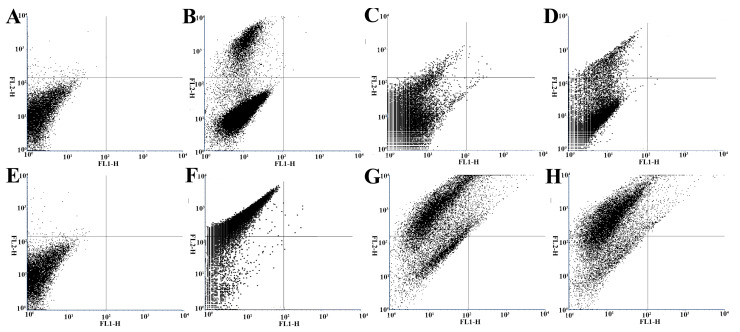
Flow cytometry images of *Trichophyton rubrum* conidia in different treatment groups at 8 h and 24 h post-treatment. (**A**,**E**) control; (**B**,**F**) MIC_TB_; (**C**,**G**) MIC_Neral_; (**D**,**H**) MIC_Geranial_. MIC, minimum inhibitory concentration.

**Figure 5 molecules-26-04263-f005:**
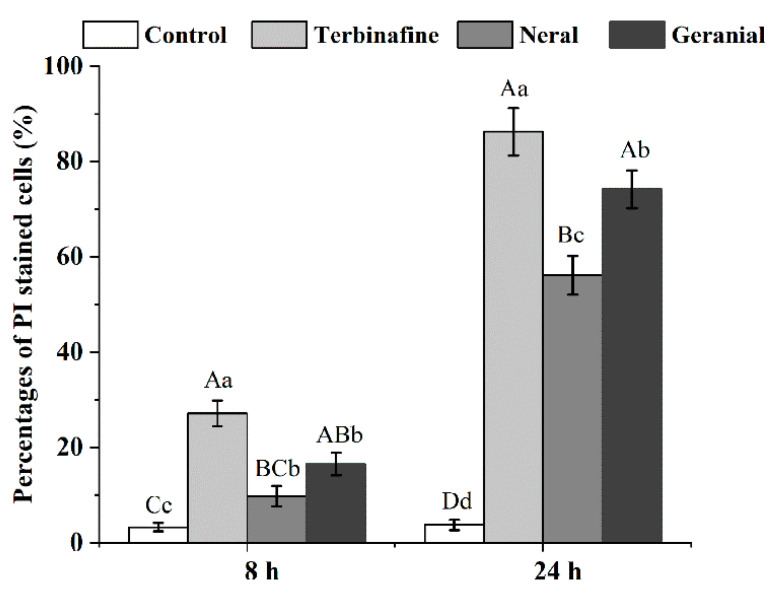
Percentages of PI-stained *Trichophyton rubrum* conidia in different treatment groups at 8 h and 24 h post-treatment. At specific time intervals, columns with lowercase or uppercase letters indicate significant or highly significant differences (*p* < 0.05 or *p* < 0.01).

**Figure 6 molecules-26-04263-f006:**
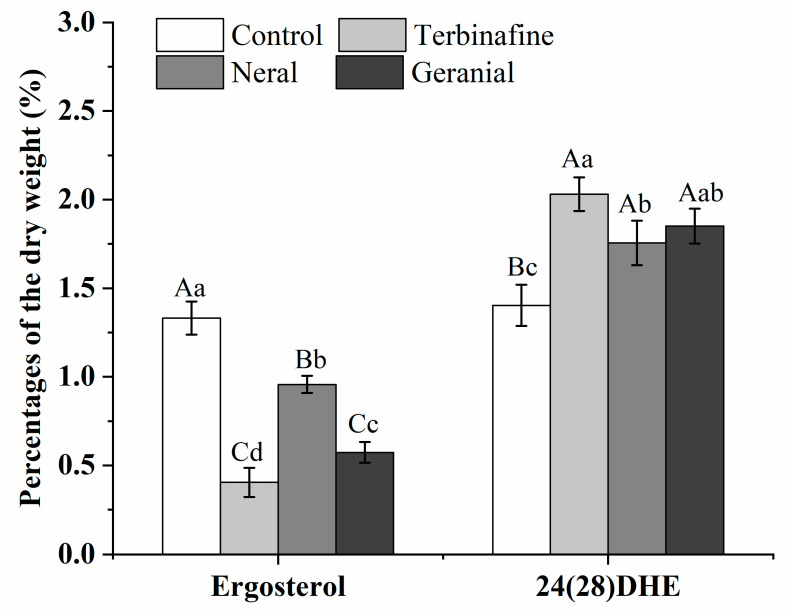
Sterol quantification of *Trichophyton rubrum* mycelium in different treatment groups at five days post-treatment. For ergosterol or 24(28)DHE, columns with lowercase or uppercase letters indicate significant or highly significant differences (*p* < 0.05 or *p* < 0.01).

**Figure 7 molecules-26-04263-f007:**
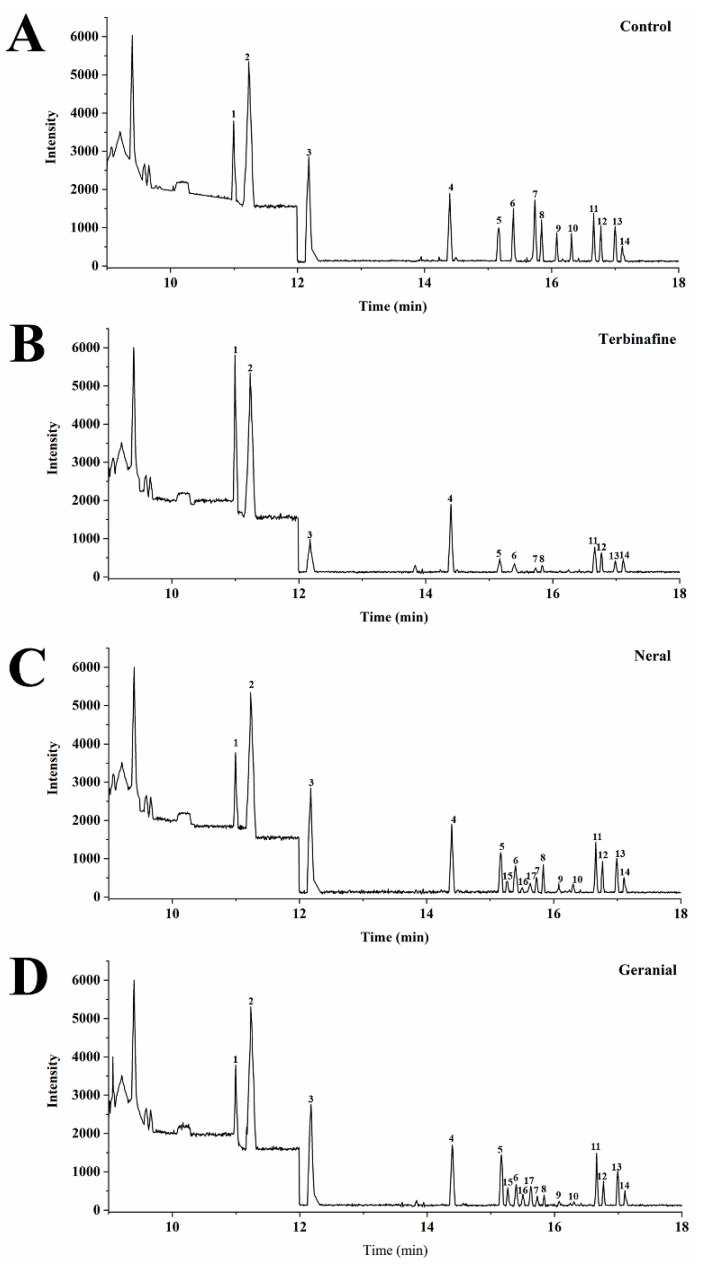
Gas chromatography diagrams of sterol patterns of *Trichophyton*
*rubrum* in different treatment groups 5 days post-treatment. (**A**) drug-free control; (**B**) terbinafine (MIC); (**C**) neral (MIC); (**D**) geranial (MIC). (1) squalene; (2) cholestane (internal standard); (3) squalene epoxide; (4) cholesterol (internal standard); (5) zymosterol; (6) ergosterol; (7) dehydroergosterol; (8) fecosterol; (9) 5-Dehydroergosterol; (10) episterol; (11) lanosterol; (12) 4-Methylfecosterol; (13) 4,4-Dimethylcholesta-8,24-dien-3β-ol(T-MAS); (14) 4,4-Dimethylcholesta-8,14,24-trien-3β-ol (FF-MAS); (15) 7-Dehydrodesmosterol; (16) Cholesta-7,24-dien-3β-ol; (17) Cholesta-5,7,22,24-tetraen-3β-ol. MIC, minimum inhibitory concentration.

**Table 1 molecules-26-04263-t001:** Inhibition zone diameter of *Trichophyton rubrum* ATCC28188 treated with specific agents.

Treatment	Inhibition Zone (mm)	
	**MIC**	**MFC**
Control	0.0 ± 0.0 ^dC^	0.0 ± 0.0 ^cC^
Terbinafine	30.2 ± 1.6 ^aA^	52.4 ± 3.0 ^aA^
Neral	20.4 ± 2.3 ^cB^	39.6 ± 2.9 ^bB^
Geranial	27.8 ± 2.5 ^bA^	50.6 ± 3.3 ^aA^

Note: results shown as mean ± standard deviation; data in line with different lowercase or capital letters indicate significant (*p* < 0.05) or highly significant (*p* < 0.01) difference. The diameter of the inhibition zone increases as the inhibitory effect increases. MIC, minimum inhibitory concentration; MFC, minimum fungicidal concentration.

**Table 2 molecules-26-04263-t002:** Sterol pattern change of ergosterol biosynthesis pathway in *Trichophyton rubrum* conidia treated with specific agents for five days.

Peak No.	Sterols	Control	Terbinafine (MIC)	Neral (MIC)	Geranial (MIC)
1	Squalene	1	2.13	0.97	0.98
2	Cholestane (IS)	1	0.97	0.98	0.98
3	Squalene epoxide	1	0.56	0.97	0.99
4	Cholesterol (IS)	1	0.98	0.95	0.99
5	Zymosterol	1	0.43	1.28	1.57
6	Ergosterol	1	0.21	0.49	0.35
7	Dehydroergosterol	1	0.07	0.22	0.14
8	Fecosterol	1	0.11	0.67	0.22
9	5-Dehydroergosterol	1	-	0.17	0.15
10	Episterol	1	-	0.18	0.15
11	Lanosterol	1	0.61	1.11	1.21
12	4-Methylfecosterol	1	0.53	0.88	0.65
13	4,4-Dimethylcholesta-8,24-dien-3β-ol (T-MAS)	1	0.25	0.97	0.98
14	4,4-Dimethylcholesta-8,14,24-trien-3β-ol (FF-MAS)	1	0.68	0.98	0.97
15	7-Dehydrodesmosterol	-	-	+	+
16	Cholesta-7,24-dien-3β-ol	-	-	+	+
17	Cholesta-5,7,22,24-tetraen-3β-ol	-	-	+	+

Note: results shown as the average folds of specific sterol amounts compared to the control; “-” represents “not detectable”; “+” represents “detectable”, but the average folds to control are unknown. MIC, minimum inhibitory concentration.

## Data Availability

All data are already provided in the manuscript.

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
