# Peer review of "Antifungal Activities of cis-trans Citral Isomers against Trichophyton rubrum with ERG6 as a Potential Target"

_molecules, 2021, doi:10.3390/molecules26144263_

Round 1
Reviewer 1 Report
The manuscript by Zheng et al shows that the antifungal activities of geranial and neral against Trichophyton rubrum (the main etiological agent of dermatophytosis).
The paper is well written and also a good design. I have few observation which can improve the text.
Line 54: It is not necessary to add this information in the introduction. This information is good to ad in the MM.
Line 224: Maybe it is more correct to call conidia instead of spore. Please, the Authors should explain more how to obtain the T. rubrum conidia. For example, what did the Authors make to to get conidia suspension without hyphae?
Reviewer 2 Report
In the section "2. Results and Discussions", the contents should be divided into several parts, which will make them more clear.
For Fig. 1, please provided the MIC and MFC concentrations of terbinafine.
Line 47: the "in vitro" should be italic.
Line 54: It should be "T. rubrum".
Line 128-129: The Aspergillus flavus should be changed to A. flavus.
The Candida albicans should be changed to C. albicans.
For Fig.4, the color figures should be provided.
Line 164: please provide the full name of DHE.
From Fig.6, we can see that the effect of citral is better than that Neral and Geranial. Please explain and discuss this result. "in contrast, the amount of late sterol intermediate 24(28) was significantly (P< 0.01, P < 0.05) increased when compared with the control.", please explain and discuss this result as well as.
Round 2
Reviewer 2 Report
The authors have undertaken an extensive revision of their manuscriptbased on my comments. They answered all the queries raised successfully.
My recommendation is to accept the revised version for publication.